# The effects of GLP-1 receptor agonists on visceral fat and liver ectopic fat in an adult population with or without diabetes and nonalcoholic fatty liver disease: A systematic review and meta-analysis

Chao Liao⊕, Xinyin Liang⊕, Xiao Zhang, Yao Li ⬮ *

Department of Endocrinology and Metabolism, The First Affiliated Hospital of Chengdu Medical College, Chengdu, China

⊕ These authors contributed equally to this work.
* njt068@163.com

**Data Availability Statement:** All relevant data are within the manuscript and its Supporting Information files.

## Abstract

### Aim

To uncover the effect of GLP-1 receptor agonists (GLP-1 RAs) on the visceral- and hepatic fat content of adults.

### Methods

PubMed, EMBASE, Cochrane Library, and Web of Science were searched from inception until November 2022. Randomized controlled trials (RCTs) of GLP-1Ras was extracted, including reports of effects on visceral adipose tissue and hepatic fat content in individuals with type 2 diabetes, non-type 2 diabetes, NAFLD (non-alcoholic fatty liver disease), and non-NAFLD. Meta-analyses used random-effects models.

### Results

1736 individuals in the 30 qualified RCTs were included, comprising 1363 people with type 2 diabetes and 318 with NFLD. GLP-1 RAs reduced visceral adipose tissue (standard mean difference [SMD] = -0.59, 95% CI [-0.83, -0.36], P<0.00001) and hepatic fat content (weighted mean difference [WMD] = -3.09, 95% CI [-4.16, -2.02], P<0.00001) compared to other control treatment. Subgroup analysis showed that GLP-1Ras dramatically decreased visceral fat in patients with type 2 diabetes (SMD = -0.49, 95% CI [-0.69, -0.29] P<0.00001), NAFLD (SMD = -0.99, 95% CI [-1.64, -0.34] P = 0.003), non-type 2 diabetes (SMD = -1.38, 95% CI [-2.44, -0.32] P = 0.01), and non-NAFLD (SMD = -0.53, 95% CI [-0.78, -0.28] P<0.0001). GLP-1Ras reduced the liver fat level of type 2 diabetes (WMD = -3.15, 95% CI [-4.14, -2.15] P<0.00001), NAFLD (WMD = -3.83, 95% CI [-6.30, -1.37] P = 0.002), and type 2 diabetes with NAFLD (WMD = -4.27, 95% CI [-6.80, -1.74] P = 0.0009), while showed no

**Funding:** The funders had no role in study design, data collection and analysis, decision to publish, or preparation of the manuscript.

**Competing interests:** The authors have declared that no competing interests exist.

impact on the hepatic fat content in non-Type 2 diabetes (WMD = −12.48, 95% CI [−45.19, 20.24] P = 0.45).

## Conclusions

LP-1 RAs significantly reduce visceral- and liver fat content in adults.

## 1 Introduction

Fat present around the abdominal viscera in the mesentery and omentum is known as visceral fat. Liver fat is one important component of visceral fat. Visceral adipose tissue can build up in arteries and has a higher metabolic rate. It can result in numerous metabolic disorders and various adipokines [1]. Visceral fat that has accumulated excessively is a major contributor to obesity. Non-esterified fatty acids, glycerol, tumor necrosis factors, leptin, resistin, and plasminogen activator inhibitor produced from adipose tissue all contribute to insulin resistance [2] and NAFLD (nonalcoholic fatty liver disease) [3] in the obese. When islet cell malfunction is combined with insulin resistance, diabetes can result [4]. Studies on obesity and diabetes have found that every kilogram of weight gained increases the risk of diabetes by 4.5 to 9% [5]. When a secondary cause of hepatic fat accumulation was excluded, steatosis >5.6% assessed by proton magnetic resonance spectroscopy or quantitative fat/water selective magnetic resonance imaging may be diagnosed as NAFLD [6]. NAFLD encompasses a broad range of diseases, from simple steatosis to nonalcoholic steatohepatitis (NASH), which can progress to fibrosis and cirrhosis [6]. Obesity has been proven to be correlated with both the incidence and prevalence of NAFLD [7]. Furthermore, obesity may be an independent predictor of liver-related mortality in NAFLD [8]. According to global estimates, 500 million people worldwide are obese, and over 1.5 billion adults are overweight [9]. Even more concerning are the rising rates of obesity among children and teenagers [10]. Incidence rates for diabetes and NAFLD in adults, which are both closely associated with obesity, are 8.8% [11] and 25% [12], respectively. These now rank among the most serious global health concerns.

As a novel class of hypoglycemic medication, GLP-1 RAs have shown some benefits in the treatment of NAFLD [13]. It also has a variety of other effects such as lowering blood pressure [14], defending the heart and brain vessels [15], lowering blood lipids, and decreasing fat accumulation. In the studies on GLP-1 Ras and visceral fat and liver fat, three recent meta-analyses showed that, for type 2 diabetes and NAFLD, regardless of the choice of exenatide or liraglutide intervention treatment, the results indicated that visceral adipose tissue and hepatic fat content were generally significantly decreased compared to other medications, placebos, and lifestyle interventions [16–18]. However, the effects of GLP-1Ras on visceral fat and liver fat content in the whole adult population (including non-NAFLD or non-type 2 diabetes) were not discussed.

Therefore, we conducted a systematic review and meta-analysis of randomized controlled trials to better understand these agents' effects on visceral fat and hepatic fat content in the adult population.

## 2 Methods

This article is a systematic review and meta-analysis of the medication effects of GLP-1 RAs in adults compared to placebos, other drugs, and active control groups. It follows the PRISMA

(preferred reporting items for systematic reviews and meta-analyses) guidelines (S1 Table) [19]. The protocol for this review was registered with PROSPERO (ID: CRD42023386822).

## 2.1 Search strategy and study selection

To find all qualifying trials, we searched the databases of PubMed, EMBASE, the Cochrane Library, and Web of Science from the earliest date available through December 1, 2022. We used terms including "('glucagon-like peptide 1' OR 'GLP-1' OR 'liraglutide' OR 'exenatide' OR 'dulaglutide' OR 'semaglutide' OR 'beinaglutide' OR 'lixisenatide') AND ('visceral fat' OR 'liver fat')" AND 'clinical trial.' Only human data from clinical trials were included in the search. We excluded those that were not randomized controlled trials (RCTs), trials for which full texts or data were unavailable, studies whose age was not matched or specified, trials with short cycles, repeated trial reports, comments, trials with no relevant outcome, and ongoing trials. The data were extracted by CL and YL, and the issue of research inclusion was addressed by consensus in extraction. The meta-analysis only included studies that complied with the following requirements: (1) the study contrasted GLP-1 RAs with placebos, other medications, or an active control group; (2) the outcomes of concern were visceral fat and hepatic fat content; (3)the standard deviation (SD) and mean (mean) estimates were presented with 95% confidence intervals (CI) (or computed data); (4) patients were all >18 years of age; (5) follow-up was done for at least four weeks.

## 2.2 Data extraction and quality assessment

The first author's last name, the publication year, the location of the study, the study design, the patient characteristics, the interventions, the diagnostic method, and the outcome measures, which included SD, mean, and 95% CI (the one that controlled for the most confounders), were all extracted from each study. The Cochrane Collaboration Risk-of-Bias tool [20] was utilized to evaluate each RCT's quality, which included sequence generation, allocation concealment, blinding, incomplete outcome data, and selective outcome reporting. The risk of bias was graded as unclear, high, or low. Inverse variance weighting and a random effects models were used to assess relative risks with 95% confidence intervals (CIs). To identify sources of heterogeneity, subgroup analyses were conducted on follow-up times, types of GLP-1Ras, body mass index (BMI), and age. Primary outcomes in this study were visceral adipose tissues and hepatic fat content. Secondary outcomes were body mass index (BMI), age, and the method used to measure outcomes.

## 2.3 Data synthesis and analysis

Revman, version 5.4 (Cochrane Collaboration), and Stata, version 16.0 (StataCorp, College Station, TX, United States) were used to analyze the data. The Mean Difference (WMD) and the standardized mean difference (SMD) were used as effect measures for continuous variables. WMD was used to quantify effect size for similar units of outcome measures, and SMD was used for the effect size of different units. Every statistical test was two-sided, and P values of <0.05 were considered significant. The measured outcome of this meta-analysis was the impact of GLP-1 RAs on adults' visceral and hepatic fat content. For efficacy outcomes, the standard deviation (SD) and 95% CI (confidence interval) were calculated. In some studies, the level of change in outcomes was not reported. We requested this information from the respective authors, or we calculated changes in findings using the conversion procedure described in Cochrane Handbook version 5.0.2. The Cochrane Q statistic was used to measure the heterogeneity between trials in the meta-analysis, and P values <0.10 indicated considerable heterogeneity [21]. The $I^2$ statistic was used to assess heterogeneity [22]. Generally, $I^2$

equal to 0% was considered homogeneous, $I^2$ values less than 25% were taken to correspond to mild heterogeneity, values between 25% and 50% were considered to have moderate heterogeneity, and values greater than 50% were taken to mean a large heterogeneity between studies. A random effects model does not account for heterogeneity. The sensitivity analysis tested the robustness of our meta-analysis results and determined whether they were biased. Subgroup analysis and meta-regression were used to explore sources of heterogeneity. The Egger Test and Begg Test were used to evaluating publication bias [23]. If needed, methods of pruning and filling were employed to evaluate the effect of publication bias [24].

## 3 Results

### 3.1 Selection of eligible studies

A total of 638 studies were found after a preliminary search of four electronic databases. After reading the titles, 206 duplicate studies were excluded. The titles and abstracts of the remaining 432 results were then carefully reviewed, and 185 studies were identified. To further determine the eligibility of each study, we reviewed the full text of these 185 papers. Of these 185 studies, 155 were ruled out for the following reasons: no result available (n = 55), unqualified or unmentioned patient ages (n = 4), ongoing trials (n = 5), no full text available (n = 31), follow-up time was less than four weeks (n = 1), non-RCTs (n = 14), no relevant outcome measures (n = 42), and commentary articles (n = 1). In the end, 30 articles [25–54] were included in this meta-analysis. The detailed screening process is shown in Fig 1.

### 3.2 Detailed characteristics of the included trials

Table 1 displays the specific characteristics of the listed studies. This analysis included 30 trials with 1420 participants, involving 1168 patients with type 2 diabetes, 234 patients with NAFLD, and 309 patients without NAFLD or type 2 diabetes. All studies included in the analysis were RCTs reported between 2009 and 2022, with sample sizes per trial ranging from 16 to 178 and follow-up periods ranging from 12 to 60 weeks. The experimental arm intervention was liraglutide in 21 studies, exenatide in seven studies, and one each for dulaglutide and semaglutide. Visceral adipose tissue (VAT) was reported in 24 studies, and hepatic fat content (HFC) was reported in 15. To measure VAT and HFC, 10 investigations utilized magnetic resonance imaging (MRI), five used computed tomography (CT), three used dual-energy X-ray absorptiometry (DXA), three used 1H-magnetic resonance spectroscopy (MRS), three used both MRI and 1H-MRS, two used MRS, two used magnetic resonance proton density fat fraction (MRI-PDFF), 1 used ultrasound (US), and 1 used a body composition analyzer BCA.

### 3.3 Risk of bias assessment

Among the 30 included articles, 22 described their precise randomization methods, eight mentioned randomization but did not specify specific randomization methods, five did not implement allocation concealment, and nine had insufficient allocation concealment methods. Sixteen publications used appropriate allocation concealment methods, nine papers used blinding for subjects, researchers, and outcome evaluators, four reports did not use blinding, three only used blinding for subjects and researchers, six did not clarify whether blinding was used, and six did not provide the reasons for dropout. All research reported expected results that were planned in the relevant Methods section. Due to their extremely low sample sizes, four studies were deemed to have a significant risk of bias. All qualified trials' overall quality levels were generally moderate. The risk of bias for each included study is summarized in Figs 2 and 3.

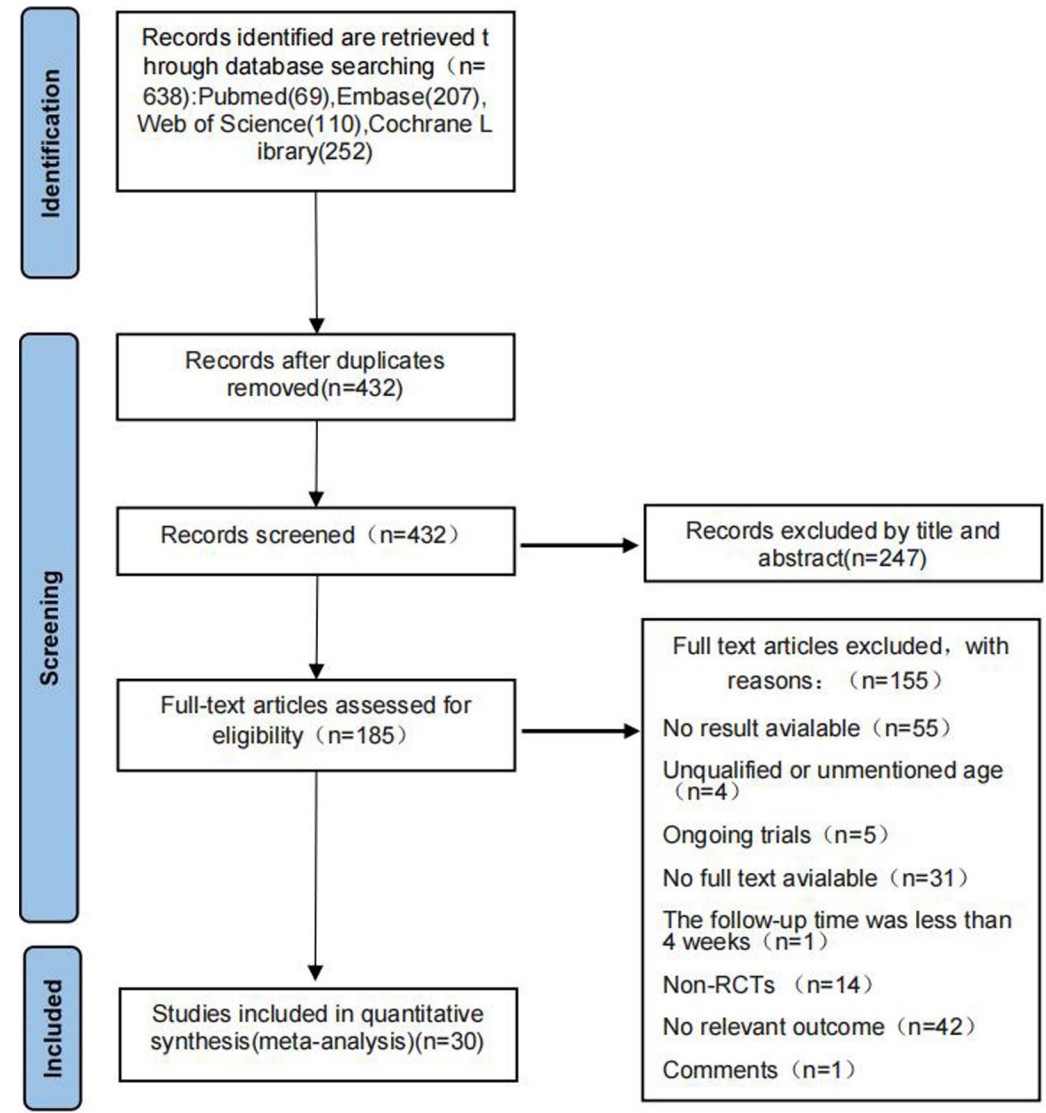

**Fig 1. PRISMA flow diagram.**

### 3.4 Meta-analysis results

**3.4.1 Visceral fat in the adult population.** There were 24 papers in total that discussed the impact of GLP-1Ras on VAT, which included a total of 1484 adult articles. The units used to measure VAT varied in each article, and SMD was used to quantify effect size. When compared to control groups such as those taking other drugs, placebo groups, and patients managing VAT by lifestyle interventions, the VAT in our GLP-1Ras group was much lower (SMD = -0.59, 95% CI [-0.83, -0.36], $I^2$ = 77%, P<0.00001; Fig 4), and the difference was statistically significant.

**3.4.2 Hepatic fat content in adults.** A total of 859 patients participated in 15 studies that examined the impact of GLP-1Ras on HFC. Each paper employed WMD as the measure of effect size, and the same unit of measurement for HFC was used throughout. The findings demonstrated that, when compared to control groups such as those taking other drugs, placebo groups, and patients managing VAT by lifestyle interventions, GLP-1RAs could considerably lower adult HFC(WMD = -3.09, 95% CI [-4.16, -2.02], $I^2$ = 40%, P<0.00001; Fig 5).

**Table 1. Characteristics of included studies.**

| Study,Year | Sample size (Experiment/ Control) | | Intervention Experimental group | Control Group | Follow-up (weeks) | Patient information Age (years) | BMI(kg/ m2) | Papulation characteristics | Outcome | Diagnostic method |
|---|---|---|---|---|---|---|---|---|---|---|
| Bi et al.2014 [25] | 11 | 11 | Exenatide 10μg BID | Pioglitazone | 24 | 52.7(1.7) | 24.5(0.5) | Type 2 Diabetes | ①VAT②HFC | 1H-MRS |
| Bizino et al.2020 [26] | 23 | 26 | Liraglutide 1.8mg QD | Placebo | 26 | 59.5(6.5) | 32.1(3.9) | Type 2 Diabetes | ①VAT | MRI and 1H-MRS |
| Bouchi et al.2017 [27] | 8 | 8 | Liraglutide 0.9 mg QD+Insulin | Insulin | 36 | 58.6 (18.9) | 27.97 (2.4) | Type 2 Diabetes | ①VAT | CT |
| Dutour et al.2016 [28] | 22 | 22 | Exenatide 10μg BID | Reference treatment | 26 | 52(1.0) | 36.1(1.1) | Type 2 Diabetes | ①VAT | MRI and 1H-MRS |
| Feng et al.2017 [29] | 29 | 58 | Liraglutide 1.8mg QD | Metformin or Gliclazide | 24 | 46.87 (11.67) | 27.4 (3.36) | Type 2 Diabetes and NAFLD | ②HFC | US |
| Frossing et al.2018 [30] | 37 | 20 | Liraglutide 1.8mg QD | Placebo | 26 | 29.9(6.1) | 33.3(4.9) | PCOS | ①VAT | MRI |
| Guo et al.2020 [31] | 31 | 60 | Liraglutide 1.8mg QD+Metformin | Insulin Glargine or Placebo +Metformin | 26 | 52.7 (6.43) | 28.8 (3.93) | Type 2 Diabetes and NAFLD | ①VAT②HFC | 1H-MRS |
| Harreiter et al.2021 [32] | 14 | 12 | Exenatide 2mg QW+Dapagliflozin 10mg QD | Placebo +Dapagliflozin 10mg QD | 24 | 60.1(7.9) | 31.3(4.1) | Type 2 Diabetes | ①VAT②HFC | MRS |
| Jendle et al.2009 [33] | 54 | 57 | Liraglutide 1.2 or 1.8mg QD +Metformin | Glimepiride or Placebo +Metformin | 26 | 18–80 | Not specified | Type 2 Diabetes | ①VAT | CT |
| Khoo et al.2019 [34] | 12 | 12 | Liraglutide 3.0mg QD | Diet and exercise interventions | 26 | 41.1(9.3) | 32.25 (3.5) | NAFLD | ②HFC | MRI |
| Kuchay et al.2020 [35] | 27 | 25 | Dulaglutide 1.5mg QW+Standard treatment | Standard treatment | 24 | 47.4 (8.96) | 29.8(3.7) | Type 2 Diabetes | ②HFC | MRI-PDFF |
| Larsen et al.2017 [36] | 47 | 50 | Liraglutide 1.8mg QD | Placebo | 16 | 42.56 (10.6) | 33.8(5.9) | Schizophrenia Spectrum | ①VAT | DXA |
| Liu et al.2020 [37] | 32 | 36 | Exenatide 10μg BID | Insulin Glargine | 24 | 49.09 (10.96) | 28.16 (3.06) | Type 2 Diabetes and NAFLD | ①VAT②HFC | MRI and 1H-MRS |
| Matikainen et al.2019 [38] | 15 | 7 | Liraglutide 1.8mg QD | Placebo | 16 | 62.3(2.0) | 32.2(3.6) | Type 2 Diabetes | ①VAT②HFC | MRI |
| McCrimmon et al.2020 [39] | 88 | 90 | Semaglutide 1.0mg QW+Canagliflozin Placebo +Metformin | Semaglutide Placebo 1.0mg QW +Canagliflozin +Metformin | 52 | 58.2 (10.0) | 32.4(5.9) | Type 2 Diabetes | ①VAT | DXA |
| Neeland et al.2021 [40] | 73 | 55 | Liraglutide 3.0mg QD | Placebo | 46 | 50.2(9.4) | 37.6(6.0) | Overweight or Obesity | ①VAT②HFC | MRI |
| Pastel et al.2017 [41] | 15 | 7 | Liraglutide 1.2mg QD | Diet interventions | 16 | 62.4(7.0) | 30.5(4.4) | Type 2 Diabetes | ①VAT | MRI |
| Salamun et al.2018 [42] | 13 | 14 | Liraglutide 1.2mg QD+Metformin | Metformin | 12 | 31.07 (4.75) | 36.7(3.5) | PCOS | ①VAT | DXA |
| Santilli et al.2017 [43] | 20 | 20 | Liraglutide 1.8mg QD | Lifestyle intervention | 60 | 53.85 (12.8) | 35.85 (8.2) | IGT,IFG, or Type 2 Diabetes | ①VAT | MRI |
| Sathyanarayana et al.2011 [44] | 11 | 10 | Exenatide 10μg BID+Pioglitazone | Pioglitazone | 48 | 52(3) | 32(1.5) | Type 2 Diabetes | ②HFC | MRS |
| Shi et al.2017 [45] | 15 | 16 | Exenatide 10μg BID | Acarbose | 12 | 41.5 (10.9) | 31.3 (2.78) | Type 2 Diabetes | ①VAT | CT |
| Smits et al.2016 [46] | 17 | 34 | Liraglutide 1.8mg QD | Sitagliptin or Placebo | 12 | 63.48 (7.1) | 31.35 (3.7) | Type 2 Diabetes | ②HFC | 1H-MRS |
| Tanaka et al.2015 [47] | 10 | 10 | Liraglutide 0.9mg QD+Metformin | Metformin | 24 | 55.0 (10.3) | 28.3(3.5) | Type 2 Diabetes | ①VAT | CT |

(*Continued*)

**Table 1.** (Continued)

| Study,Year | Sample size | | Intervention | | Follow-up (weeks) | Patient information | | Papulation characteristics | Outcome | Diagnostic method |
|---|---|---|---|---|---|---|---|---|---|---|
| | (Experiment/ Control) | | Experimental group | Control Group | | Age (years) | BMI(kg/ m2) | | | |
| Tang et al.2015 [48] | 18 | 17 | Liraglutide 1.8mg QD | Insulin Glargine | 12 | 60.55 (12.89) | 31.25 (4.5) | Type 2 Diabetes | ②HFC | MRI |
| Vanderheiden et al.2016 [49] | 32 | 34 | Liraglutide 1.8mg QD+Insulin | Placebo+Insulin | 24 | 54.15 (7.35) | 41.15 (8.55) | Type 2 Diabetes | ①VAT②HFC | MRI |
| Van Eyk et al.2019 [50] | 22 | 25 | Liraglutide 1.8mg QD | Placebo | 26 | 55.0(10, 0) | 29.5(3.9) | Type 2 Diabetes | ①VAT | MRI |
| Wang et al.2020 [51] | 40 | 41 | Exenatide 10μg BID | humalog mix 25 | 26 | 58.23 (10.98) | 23.73 (1.21) | Type 2 Diabetes | ①VAT②HFC | MRI |
| Yan et al.2019 [52] | 24 | 51 | Liraglutide 1.8mg QD+Metformin | Insulin Glargine or Sitagliptin +Metformin | 26 | 44.4 (9.02) | 29.88 (3.18) | Type 2 Diabetes and NAFLD | ①VAT②HFC | MRI-PDFF |
| Yu et al.2022 [53] | 47 | 38 | Liraglutide | Lifestyle intervention | 12 | 18–65 | 31.8(4.0) | Type 2 Diabetes | ①VAT | CT |
| Yu DN et al.2020 [54] | 39 | 24 | Liraglutide 1.8mg QD | Diet | 12 | 52.6(9.8) | 31.8 (4.97) | Type 2 Diabetes | ①VAT | BCA |

Note: Values are expressed as means ± standard deviation; QD: 1 time per day; BID:2 times per day; QW: once a week; Reference treatment: Bedtime insulin glargine plus oral glimepiride or other sulfonylureas; Standard treatment: Insulin combined with metformin, sulfonylureas, DPP4 inhibitor therapy; BMI: body mass index; PCOS: polycystic ovarian syndrome; NAFLD: Non-Alcoholic Fatty Liver Disease; VAT: Visceral adipose tissue; HFC: Hepatic fat content; 1H-MRS:1H-magnetic resonance spectroscopy; MRS: magnetic resonance spectroscopy; MRI: magnetic resonance imaging; MRI-PDFF: magnetic resonance proton density fat fraction; CT: computed tomography; DXA: dual-energy x-ray absorptiometry; US: ultrasound; BCA: body composition analyzer.

**3.4.3 Visceral fat in different population groups.** Of the 24 studies reporting the effect of GLP-1Ras on visceral fat, 19 studies had participants with type 2 diabetes, three with NAFLD, four without type 2 diabetes, and 21 with non-NAFLD. Compared to the control group, GLP-1Ras significantly reduced visceral fat in patients with type 2 diabetes (SMD = -0.49, 95% CI [-0.69, -0.29], $I^2$ = 57%, P<0.00001; Fig 6), NAFLD patients (SMD = -0.99, 95% CI [-1.64, -0.34], $I^2$ = 81%, P = 0.003; Fig 6), non-type 2 diabetes (SMD = -1.38, 95% CI [-2.44, -0.32], $I^2$ = 93%, P = 0.01; Fig 6), and non-NAFLD populations (SMD = -0.53, 95% CI [-0.78, -0.28], $I^2$ = 75%, P<0.0001; Fig 6). The difference in the individual item analysis results was statistically significant.

**3.4.4 Hepatic fat content in different population groups.** The 15 trials that reported the effect of GLP-1 on liver fat content included 13 trials in persons with type 2 diabetes, five trials in patients with NAFLD, two trials in patients without type 2 diabetes, and 10 trials in patients without NAFLD. Of the five trials with NAFLD, four included patients with both type 2 diabetes and NAFLD. GLP-1Ras was found to substantially lower the HFC of type 2 diabetes patients (WMD = -3.15, 95% CI [-4.14, -2.15], $I^2$ = 35%, P<0.00001; Fig 7) and the non-NAFLD population (WMD = -2.81, 95% CI [-4.04, -1.58], $I^2$ = 34%, P<0.00001; Fig 7) compared to the control group. The impact on those with NAFLD (WMD = -3.83, 95% CI [-6.30, -1.37], $I^2$ = 55%, P = 0.002; Fig 7), particularly those who also have type 2 diabetes (WMD = -4.27, 95% CI [-6.80, -1.74], $I^2$ = 58%, P = 0.0009; Fig 7), was also found to be substantial. The results of these analyses were statistically significant. However, for non-type 2 diabetes, the analysis results showed that GLP-1ra may also lower HFC (WMD = -12.48, 95% CI [-45.19, 20.24], I2 = 77%, P = 0.45; Fig 7). The change, however, was not statistically significant.

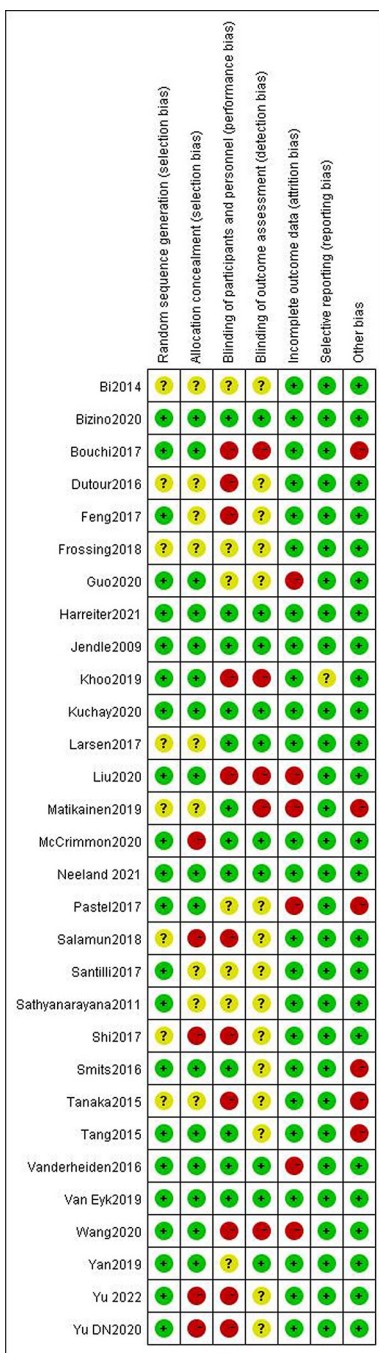

**Fig 2. Summary of risk of bias for included articles: Review the authors' judgements for each included study's risk of bias.**

### 3.5 Sensitivity analysis, subgroup analysis, and meta-regression of VAT and HFC

Given that the combined effect for VAT and HFC showed varied degrees of heterogeneity, with $I^2$ = 34%~93%, sensitivity analyzes were performed and showed that the results of each analysis were stable (S1-S11 Fig in S1 File). We then conducted subgroup analysis on the

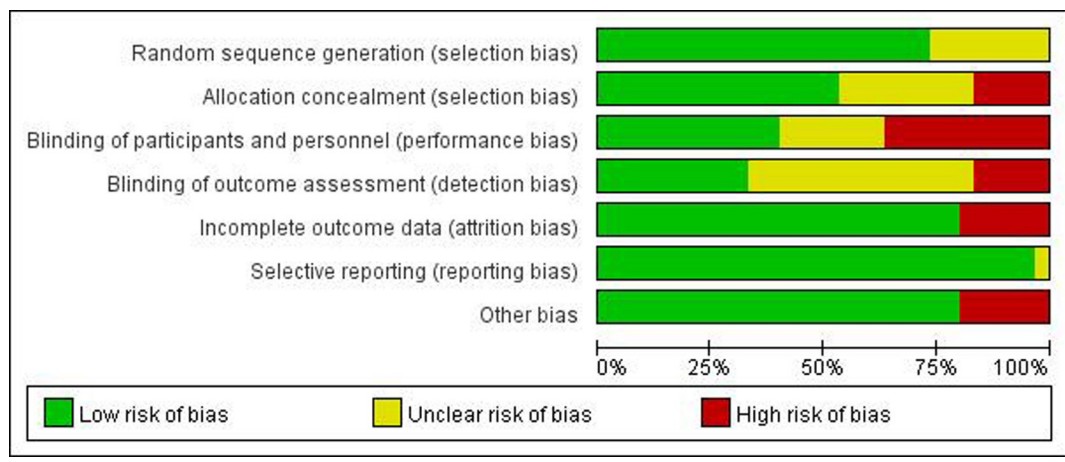

**Fig 3. Inclusion article risk of bias map: Review authors' judgments of item risk of bias in all included studies, expressed as a percentage.**

follow-up times, the type of GLP-1Ras, body mass index (BMI), age, and the method used to measure outcomes, in an attempt to elucidate sources of heterogeneity. The analysis results are displayed in Table 2.

With regard to VAT, subgroup analysis based on GLP-1 type revealed that for patients taking liraglutide (SMD = -0.68, 95% CI [-1.00, -0.36]); peptides (SMD = -0.46, 95% CI [-0.70, -0.21]); and other types of GLP-1Ra (SMD = -0.68, 95% CI [-1.00, -0.36]), the P and $I^2$ values between the groups were 0.06 and 63.7%, respectively (S12 Fig in S1 File). Other subgroup analyses, such as those based on BMI (S13 Fig in S1 File), age (S14 Fig in S1 File), outcome measuring methods (S15 Fig in S1 File), and follow-up time (S16 Fig in S1 File), had P values greater than 0.1 and $I^2$ values less than 50%. For HFC, the subgroup analysis based on the outcomes measurement method showed that for the subgroup using magnetic resonance class

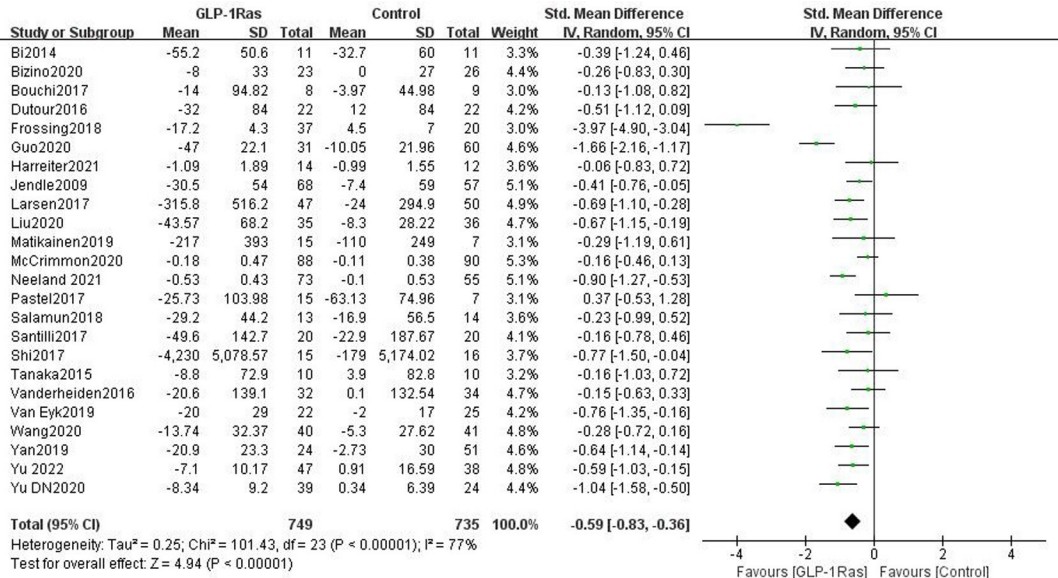

**Fig 4. Forest plot comparing the post-treatment visceral adipose tissue (VAT) of the control and GLP-1Ras group.**

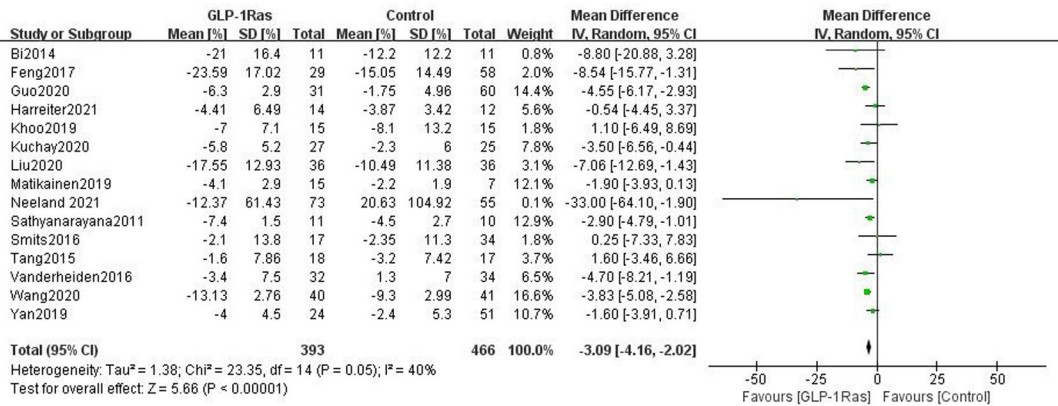

**Fig 5. Forest plot comparing post-treatment hepatic fat content (HFC) of the control and GLP-1Ras group.**

(WMD = -2.99, 95%CI [-4.04,-1.94]) and the subgroup using other measurement methods (WMD = -8.54, 95%CI [-15.77, -1.31]), the $I^2$ between the groups was 54.9% and the P value was 0.14 (S20 Fig in S1 File). Other subgroup analyses including follow-up time (S16 Fig in S1 File), GLP-1Ra type (S17 Fig in S1 File), BMI (S18 Fig in S1 File), and age (S19 Fig in S1 File), all had P values greater than 0.1, and between groups all had $I^2$ values below 50%. The results of our subgroup analysis indicated that the type of GLP-1Ras for VAT and the method of outcome measurement for HFC may both have been sources of heterogeneity.

Based on these two variables, we performed a meta-regression analysis. The results revealed that there was no significant link between VAT and the regression analysis of GLP-1Ras type P = 0.388 (S21 Fig in S1 File). In terms of HFC, the regression analysis of outcome measurement method P = 0.232 showed no significant link with HFC (S22 Fig in S1 File).

### 3.6 Assessment of publication bias

For the two outcome indicators of VAT and HFC, we made funnel plots, which were found to be visually symmetrical (Figs 8 and 9). In order to check for publication bias, we ran Egger's test and Begg's test. Egger's test for VAT indicated a P = 0.537, and Begg's test had a P = 0.673. Regarding HFC, Egger's test yielded P = 0.900 and Begg's test P = 0.621. We therefore showed that publication bias was low.

## 4. Discussion

This study represents the first systematic analysis of the impact of GLP-1Ras on visceral obesity and hepatic ectopic fat in the adult population. A total of 30 RCTs with 1738 individuals were included. In comparison to placebos, other drugs, or an active control group, GLP-1RA treatment was shown to substantially reduce VAT and HFC in the whole adult population, according to our findings. In terms of HFC, GLP-1Ras can considerably lower the HFC of a variety of populations, including those with type 2 diabetes, NAFLD (including those with type 2 diabetes and NAFLD), and non-NAFLD populations.

Much research has confirmed that excess visceral fat not only causes type 2 diabetes [55], insulin resistance [3], and inflammatory disorders [56], but is also connected with the occurrence of cardiometabolic diseases [57]. According to a study conducted by the University of Tokyo School of Medicine on the relationship between visceral fat and atherosclerosis indicators, increased visceral fat was an independent risk factor for higher brachial-ankle pulse wave velocity [58]. Oikawa et al. discovered that the percentage of visceral fat as body weight was an

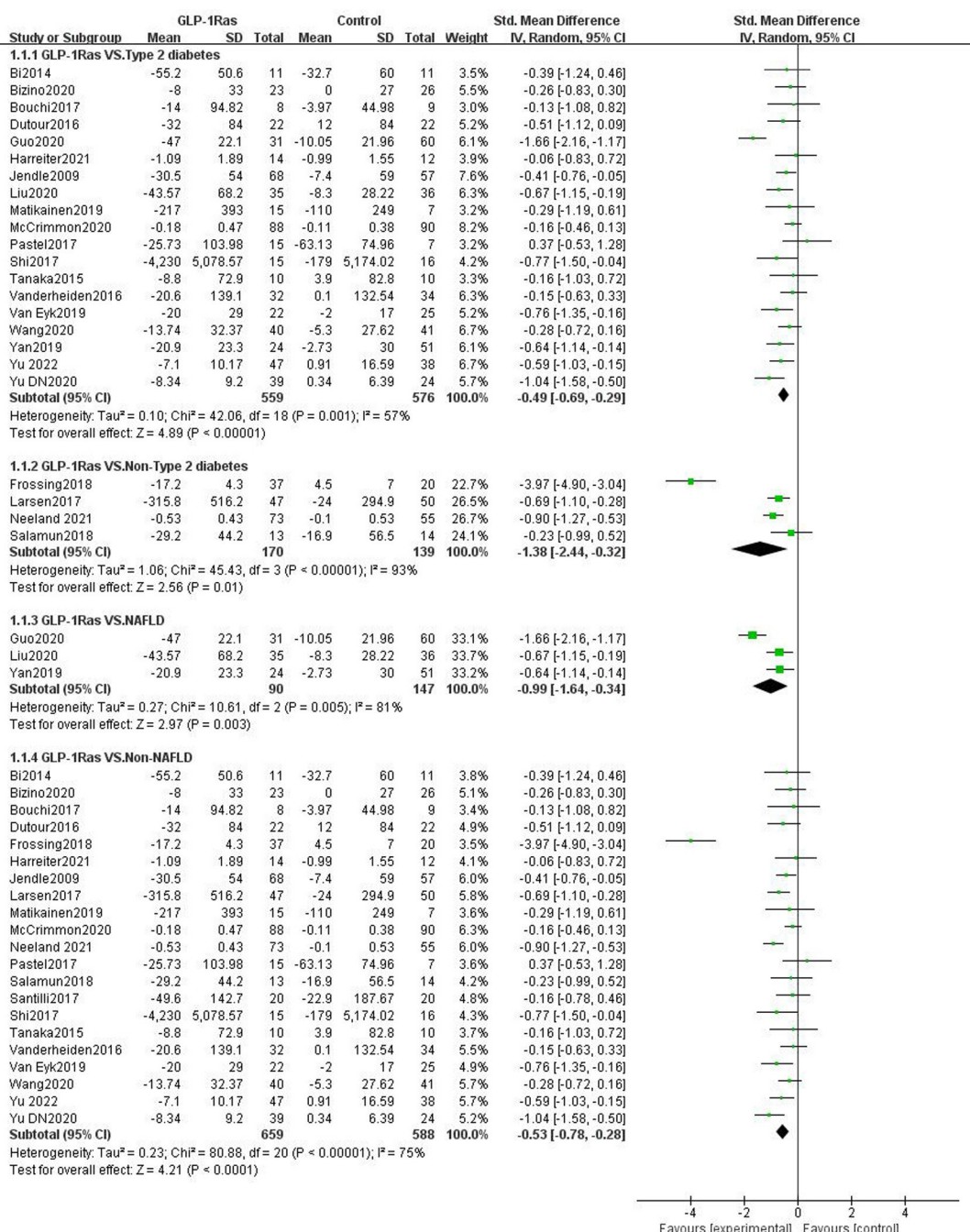

**Fig 6. Forest plots comparing post-treatment visceral adipose tissue (VAT) in the control and GLP-1Ras groups of different populations.**

independent risk factor for aortic valve calcification [59]. Moreover, abdominal obesity caused by increased abdominal visceral fat is a sign of cardiovascular disease and metabolic risk [60]. Visceral fat is also linked to the occurrence and progression of many malignancies, such as lung cancer [61], colorectal adenomas [62, 63], gastric cancer [64, 65]. Hepatic fat is a risk factor of NAFLD patients to develop diabetes mellitus, metabolic syndrome, and high blood pressure [66, 67].

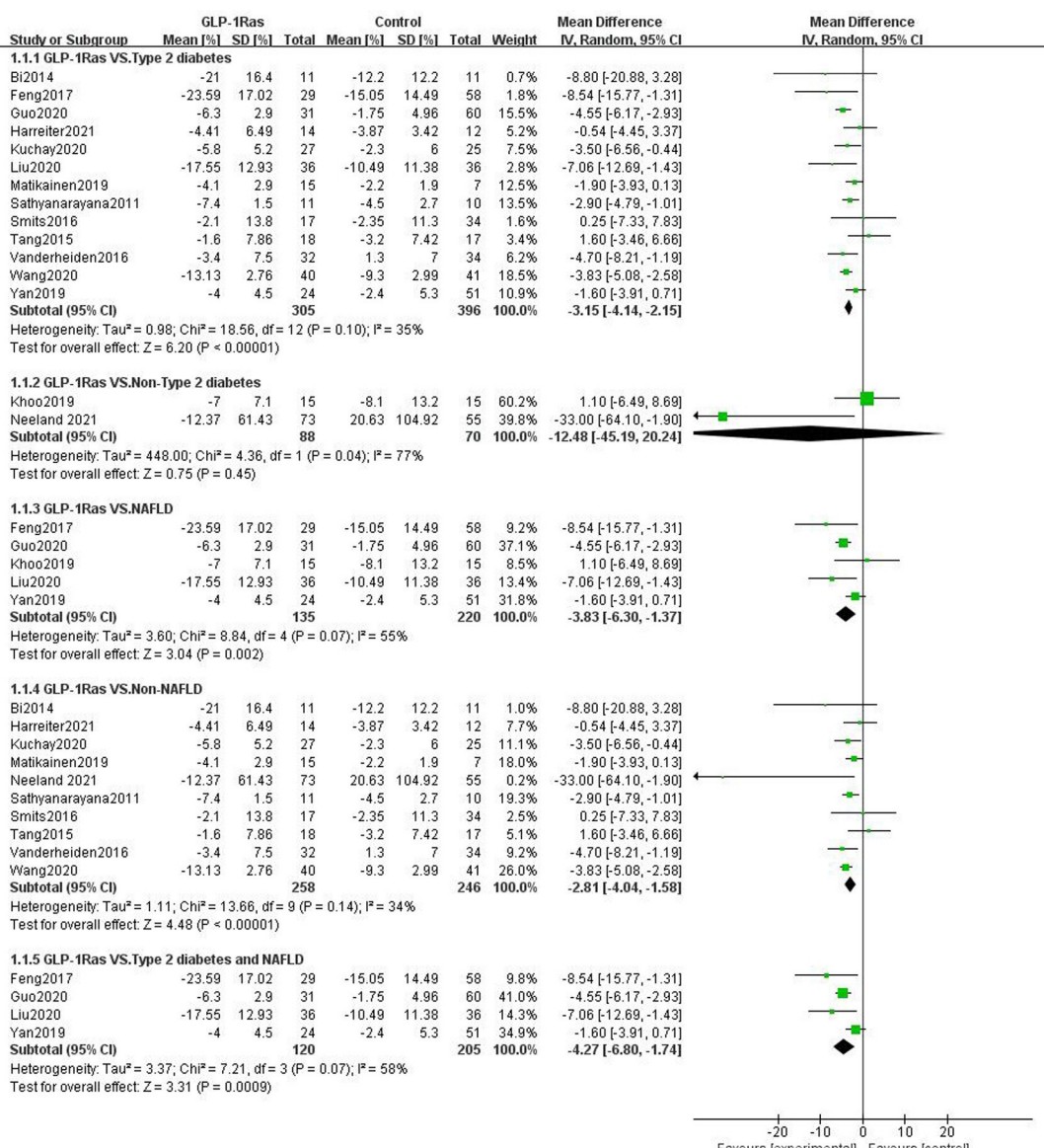

**Fig 7. Forest plots comparing post-treatment hepatic fat content(HFC) in thecontrol and GLP-1Ras groups of different populations.**

Therefore, lowering visceral fat, including liver fat, has become one of the treatment aims for metabolic disorders linked to obesity, since it is detrimental to the human body and is one of the causes of the occurrence and development of numerous diseases. As of right now, the list of approved anti-obesity drugs includes sympathomimetic medications such as diethylpropion [68], afepramone [68], phenmetrazine [68], phendimetrazine [68], cathine (nor-pseudo-ephedrine) [69], phentermine [70], pancreatic lipase inhibitor such as orlistat [71], anticonvulsants such as phentermine or topiramate [72, 73], and opioid receptor antagonists or dopamine and noradrenaline reuptake inhibitors such as naltrexone [74] and bupropion [74]. All of these exhibit varying degrees of negative effects and side effects, some of which include cardiovascular adverse effects, higher risk of suicide, or greater possibility for misuse dependence. Some medications are only recommended for short-term use due to the potential

**Table 2. Subgroup analysis of the effects of glucagon-like peptide-1 receptor agonists on visceral adipose tissue and hepatic fat content.**

| Grouping characteristics | Study number | VAT SMD(95%CI) | I2 between groups | P | Study number | HFC(%) WMD(95%CI) | I2 between groups | P |
|---|---|---|---|---|---|---|---|---|
| Follow-up time | | | | | | | | |
| >24weeks | 12 | -0.77(-1.17,-0.37) | 35.40% | 0.21 | 6 | -3.30(-4.65,-1.96) | 0% | 0.73 |
| ≤24weeks | 12 | -0.59(-0.83,-0.36) | | | 9 | -2.90(-4.16,-2.02) | | |
| GLP-1Ras type | | | | | | | | |
| Liraglutide | 17 | -0.68(-1.00,-0.36) | 63.70% | 0.06 | 9 | -2.71(-4.59,-0.83) | 0% | 0.82 |
| Exenatide | 6 | -0.46(-0.70,-0.21) | | | 5 | -3.42(-4.77,-2.06) | | |
| Other GLP-1Ra | 1 | -0.16(-0.46,-0.13) | | | 1 | -3.50(-6.56,-0.44) | | |
| Baseline BMI | | | | | | | | |
| >30 kg/m2 | 15 | -0.60(-0.93,-0.26) | 14.50% | 0.31 | 8 | -1.99(-3.64,-0.33) | 46.20% | 0.16 |
| 25–30 kg/m2 | 6 | -0.74(-1.19,-0.30) | | | 5 | -3.96(-5.83,-2.09) | | |
| <25kg/m2 | 2 | -0.30(-0.69,-0.09) | | | 2 | -3.88(-5.13,-2.64) | | |
| Age | | | | | | | | |
| >50 years old | 16 | -0.46(-0.71,-0.20) | 11.50% | 0.32 | 10 | -3.05(-4.29,-1.81) | 0% | 0.8 |
| 40–50 years old | 4 | -0.68(-0.93,-0.44) | | | 5 | -3.41(-5.96,-0.86) | | |
| <40 years old | 2 | -2.09(-5.76,1.57) | | | | | | |
| Outcome Measures Method | | | | | | | | |
| Magnetic resonance class | 15 | -0.67(-1.04,-0.30) | 38.50% | 0.18 | 14 | -2.99(-4.04,-1.94) | 54.90% | 0.14 |
| CT | 5 | -0.47(-0.71,-0.23) | | | | | | |
| DXA | 3 | -0.37(-0.75,0.01) | | | | | | |
| Other measurement methods | 1 | -1.04(-1.58,-0.50) | | | 1 | -8.54(-15.77,-1.31) | | |

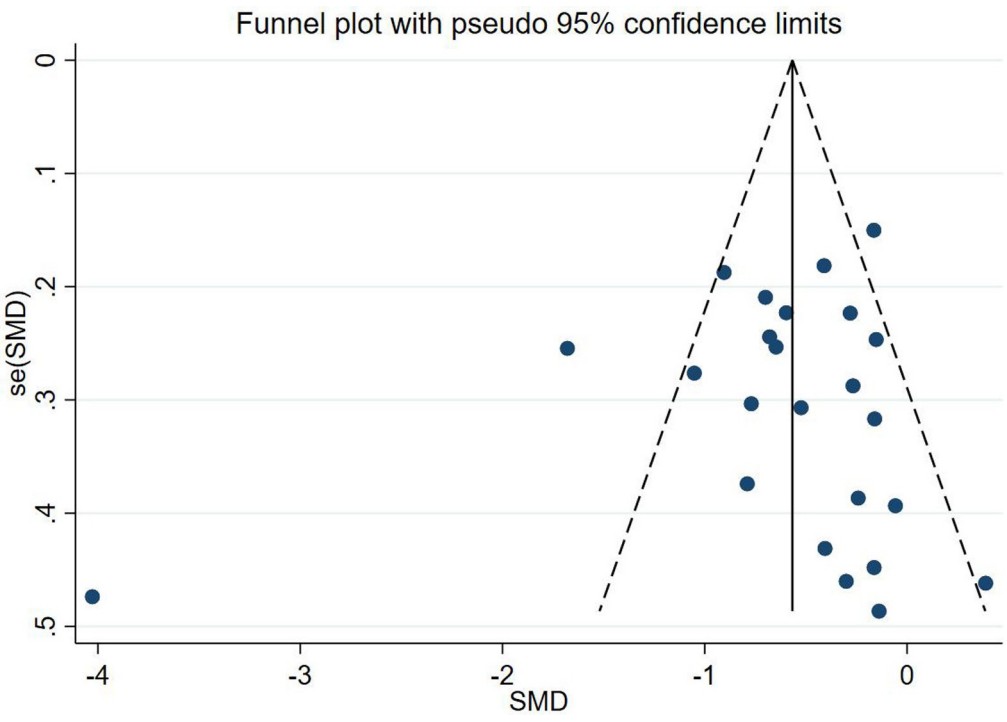

**Fig 8. Funnel plot of VAT.**

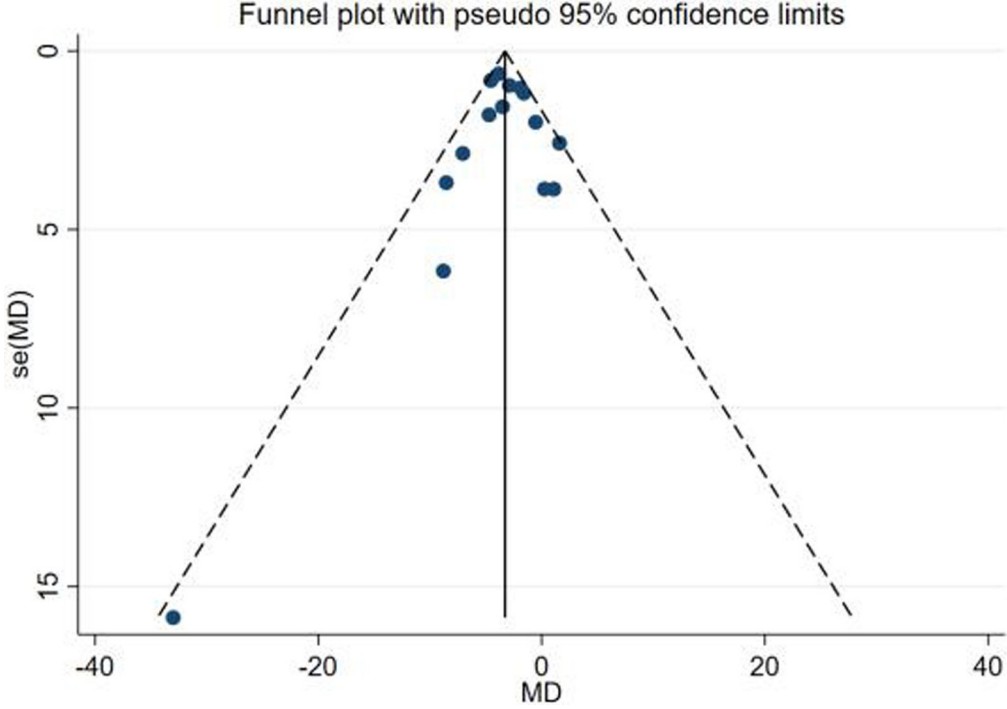

**Fig 9. Funnel plot of HFC.**

for addiction or allergic reactions. For NAFLD, there are presently no pharmacological medications approved to treat the condition specifically, with weight loss being the only recognized therapeutic option [75].

An increasing body of research has revealed that GLP-1 can affect the central nervous system by acting on locations such as the hypothalamic feeding center, the hindbrain matrix, the hindbrain nucleus, parabrachial lateral nucleus, and the mesencephalic limbic stroma [76–80]. It can enhance satiety and decreases food intake through related nerve conduction. GLP-1 is also broadly expressed in the gastrointestinal and nervous systems, where it inhibits gastric emptying and intestinal motility effects [81]. Additionally, it can prevent the release of chylomicrons after meals [78], lower blood triglyceride levels, and, after attaching to receptors on liver cells, it can also lessen hepatic steatosis levels [79]. GLP-1 can directly act on adipose tissue, reduce the synthesis of white adipose tissue, and reduce the thickness of visceral fat [82]. The emergence of GLP-1Ra drugs provides us with another possibility for long-term drug treatment to normalize body weight, visceral fat, and hepatic ectopic fat. In the United States, GLP-1 RAs such as liraglutide and semaglutide have been licensed by the US Food and Drug Administration (FDA) for the treatment of obesity and excessive weight, regardless of whether the patient has diabetes.

In this meta-analysis, the heterogeneity of VAT and HFC were relatively large after the studies were combined. In order to clarify the source of this heterogeneity, we performed a subgroup analysis and a sensitivity analysis. Our sensitivity analysis suggested that the results of each analysis were stable. Our subgroup analysis based on five parameters: follow-up times, GLP-1Ras type, BMI, age, and outcome measuring method, showed that the type of GLP-1Ras and outcome measuring method may have been the main sources of heterogeneity.

To further clarify the source of heterogeneity, we conducted a meta-regression analysis on the above two factors. Both the type of GLP-1Ras and visceral fat had no significant link,

according to the results, and neither did HFC or the way the results were measured. These two factors therefore were not potential sources of heterogeneity. The source of heterogeneity may still be present in other aspects, therefore, which need to be investigated further.

Three recent meta-analyses showed that GLP-1Ras treatment results in the significant reduction of visceral adipose tissue and hepatic fat content in the patients with type 2 diabetes and NAFLD compared with other medications, placebos, and lifestyle interventions [16–18]. Consistently, our meta-analysis suggests a possible beneficial class effect of GLP-1 RAs on serum liver enzyme levels and imaging-detected liver fat content of patients with NAFLD and type 2 diabetes. Our meta-analysis provide a broader representation of important clinical outcomes of GLP-1Ras on visceral fat and liver fat content in the whole adult population (including non-NAFLD or non-type 2 diabetes). Additionally, as compared with previously published reviews, our systematic review included a much larger number of trials and patients.

This study has some key limitations. First, all of the included studies were written in English, which could have resulted in publication or selection bias. Second, quite a few of the included studies were not double-blind studies, and many of them received funding from pharmaceutical companies, which would have produced more positive efficacy findings and conclusions. Third, not all of the chosen studies were placebo-controlled, potentially increasing heterogeneity. Fourth, part of the results of this study showed high heterogeneity, which needs to be clarified by further research. Although we conducted meta-regression and subgroup analysis, no distinct causes of heterogeneity were found. Fifth, there are limited data regarding effects on HFC for the non-NAFLD population, which may have led to instabilities in the results. Further well-designed tests are therefore needed to validate the GLP effects of 1Ras on HFC in non-NAFLD population. Finally, due to the limits of the relevant literature, the GLP-1 RAs included in this analysis only included liraglutide, exenatide, dulaglutide, and semaglutide, with a heavier focus on liraglutide and exenatide, so we were not able to explore the effects of all forms of GLP-1Ras on VAT and HFC. We also did not evaluate the effect of GLP-1Ras on other ectopic fat distributions, such as myocardial fat, perivascular fat, pericardial fat, intra-myocardial fat, epicardial fat, cardiac fat, pancreatic fat. Larger RCTs are needed to corroborate all of the findings in this analysis.

The present meta-analysis has clear clinical implications. Based on our results, GLP-1Ras has significantly reduced both VAT and HFC in adults. This indicates that GLP-1Ras could be a promising treatment option for type 2 diabetes, NAFLD, non-type 2 diabetes or non-NAFLD patients who have high levels of VAT and HFC.

## 5. Conclusion

In conclusion, GLP-1Ras can drastically lower visceral fat and HFC in adults. The results have important implications for clinical practice. We found that GLP-1Ras can reduce visceral fat and liver fat content in patients with both type 2 diabetes and NAFLD suggesting that GLP-1Ras should be more widely used in the treatment of these conditions. GLP-1Ras could be recommend as an alternative treatment for various metabolic diseases that cause visceral fat accumulation. Additionally, GLP-1Ras also has the potential to become a specific therapy for NAFLD.

## Supporting information

**S1 Table. The current study follows the PRISMA (preferred reporting items for systematic reviews and meta-analyses) guidelines.**
(DOCX)

**S1 File. Sensitivity analysis charts, forest plots, and meta-regression analysis in this study.** (DOCX)

## Author Contributions

**Conceptualization:** Yao Li.

**Data curation:** Xiao Zhang.

**Formal analysis:** Xiao Zhang.

**Methodology:** Chao Liao.

**Resources:** Xiao Zhang.

**Writing – original draft:** Chao Liao, Xinyin Liang.

**Writing – review & editing:** Yao Li.

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
