## [Decision Letter · Decision Letter 0]

20 Mar 2023

PONE-D-23-05315The effects of GLP-1 receptor agonists on visceral fat and liver ectopic fat in an adult population with or without diabetes and nonalcoholic fatty liver disease: a systematic review and meta-analysisPLOS ONE

Dear Dr. Li,

Thank you for submitting your manuscript to PLOS ONE. After careful consideration, we feel that it has merit but does not fully meet PLOS ONE’s publication criteria as it currently stands. Therefore, we invite you to submit a revised version of the manuscript that addresses the points raised during the review process.

We look forward to receiving your revised manuscript.

Kind regards,

Tariq Jamal Siddiqi

Academic Editor

PLOS ONE

Journal Requirements:

Reviewers' comments:

Reviewer's Responses to Questions

**Comments to the Author**

1. Is the manuscript technically sound, and do the data support the conclusions?

Reviewer #1: Yes

2. Has the statistical analysis been performed appropriately and rigorously? 

Reviewer #1: Yes

3. Have the authors made all data underlying the findings in their manuscript fully available?

Reviewer #1: Yes

4. Is the manuscript presented in an intelligible fashion and written in standard English?

Reviewer #1: Yes

5. Review Comments to the Author

Reviewer #1: Liao et al has conducted a meta-analysis on “The effects of GLP-1 receptor agonists on visceral fat and liver ectopic fat in an adult population with or without diabetes and nonalcoholic fatty liver disease” and concluded that e GLP-1 receptor agonists significantly reduce visceral and liver fat content in adults. These findings were especially true for patients with both type 2 diabetes and NAFLD, hence suggest that GLP-1Ras should be more widely used in the treatment of these conditions. This can prove to be valuable finding, however, in my opinion, the manuscript can be improved by incorporating the following edits:

1. The abstract exceeds the word limit of 250 words and is very lengthy. The authors should consider summarizing the content and providing only the main findings of the study in the abstract.

2. In introduction, the authors should use expanded form of NAFLD it was used here the first time in the manuscript other than the abstract.

3. In 2nd paragraph of the ‘introduction’, the authors should write the abbreviation of GLP-1 receptor agonists as ‘GLP-1 RAs’. Moreover, the authors have used ‘GLP-1 Ras’ as the abbreviation elsewhere in the manuscript, which should be corrected.

4. In the ‘data extraction and quality assessment’ section, the authors should explain all the outcomes, primary and secondary, and specify subgroups for better comprehensibility of the manuscript.

5. The numbering of the sections of the results is not correct. The authors should correct the numbering of the subheadings to avoid confusion in the readers.

6. In subsection ‘Visceral fat in different population groups’ of the results, the authors have stated that “Of the 24 studies reporting the effect of GLP-1Ras on visceral fat, 19 studies had participants with type 2 diabetes, three with NAFLD, four without type 2 diabetes, and 21 with type 2 non-NAFLD.” The authors should correct the part ‘type 2 non-NAFLD’ by removing ‘type 2’ to make better sense of the information.

7. For all outcomes in the ‘results’, provide a reference for the forest plot showing the result of that outcome.

8. The authors should make a comparison of this meta-analysis with previously existing studies to highlight the significance of this study and how it adds to the current literature.

9. 2nd, 3rd and 4th paragraphs of the discussion provide a very detailed content of the number of conditions predisposed by obesity. However, this information is not in relevance with the scope of this study in as much detail. This information could be summarized into few lines and linked to clinical implications of the findings of this study. This would help maintain relevance and coherence.

10. The authors should consider providing a distinct paragraph in the discussion for ‘clinical implications’ to highlight the significance of the findings in clinical practice.

6. PLOS authors have the option to publish the peer review history of their article (what does this mean?). If published, this will include your full peer review and any attached files.

Reviewer #1: No

---

## [Author Response · Author response to Decision Letter 0]

4 Apr 2023

Journal Requirements:

Reviewers' comments:

Reviewer's Responses to Questions

Comments to the Author

1. Is the manuscript technically sound, and do the data support the conclusions?

Reviewer #1: Yes

2. Has the statistical analysis been performed appropriately and rigorously?

Reviewer #1: Yes

3. Have the authors made all data underlying the findings in their manuscript fully available?

Reviewer #1: Yes

4. Is the manuscript presented in an intelligible fashion and written in standard English?

Reviewer #1: Yes

5. Review Comments to the Author

Reviewer #1: Liao et al has conducted a meta-analysis on “The effects of GLP-1 receptor agonists on visceral fat and liver ectopic fat in an adult population with or without diabetes and nonalcoholic fatty liver disease” and concluded that e GLP-1 receptor agonists significantly reduce visceral and liver fat content in adults. These findings were especially true for patients with both type 2 diabetes and NAFLD, hence suggest that GLP-1Ras should be more widely used in the treatment of these conditions. This can prove to be valuable finding, however, in my opinion, the manuscript can be improved by incorporating the following edits:

1. The abstract exceeds the word limit of 250 words and is very lengthy. The authors should consider summarizing the content and providing only the main findings of the study in the abstract.

Response: Thank you for your helpful comments. We shortened the abstract accordingly.

2. In introduction, the authors should use expanded form of NAFLD it was used here the first time in the manuscript other than the abstract.

Response: Thank you for your helpful comments. We expanded the NAFLD in the first use in the introduction.

3. In 2nd paragraph of the ‘introduction’, the authors should write the abbreviation of GLP-1 receptor agonists as ‘GLP-1 RAs’. Moreover, the authors have used ‘GLP-1 Ras’ as the abbreviation elsewhere in the manuscript, which should be corrected.

Response: Thank you for your critical comments. We revised it accordingly.

4. In the ‘data extraction and quality assessment’ section, the authors should explain all the outcomes, primary and secondary, and specify subgroups for better comprehensibility of the manuscript.

Response: Thank you for your helpful comments. We detailed the primary and secondary outcomes and specified the subgroups.

5. The numbering of the sections of the results is not correct. The authors should correct the numbering of the subheadings to avoid confusion in the readers.

Response: Thank you for your critical comments. We remumbered the section of the results.

6. In subsection ‘Visceral fat in different population groups’ of the results, the authors have stated that “Of the 24 studies reporting the effect of GLP-1Ras on visceral fat, 19 studies had participants with type 2 diabetes, three with NAFLD, four without type 2 diabetes, and 21 with type 2 non-NAFLD.” The authors should correct the part ‘type 2 non-NAFLD’ by removing ‘type 2’ to make better sense of the information.

Response: Thank you for your helpful comments. We revised it accordingly.

7. For all outcomes in the ‘results’, provide a reference for the forest plot showing the result of that outcome.

Response: Thank you for your helpful comments. We provided a reference for each forest plot in the result section.

8. The authors should make a comparison of this meta-analysis with previously existing studies to highlight the significance of this study and how it adds to the current literature.

Response: Thank you for your critical comments. We compared our work with previously existing studies.

9. 2nd, 3rd and 4th paragraphs of the discussion provide a very detailed content of the number of conditions predisposed by obesity. However, this information is not in relevance with the scope of this study in as much detail. This information could be summarized into few lines and linked to clinical implications of the findings of this study. This would help maintain relevance and coherence.

Response: Thank you for your insightful comments. We rewrote these detailed content of the number of conditions predisposed by obesity accordingly.

10. The authors should consider providing a distinct paragraph in the discussion for ‘clinical implications’ to highlight the significance of the findings in clinical practice.

Response: Thank you for your insightful comments. We added the clinical implications in the discussion.

6. PLOS authors have the option to publish the peer review history of their article (what does this mean?). If published, this will include your full peer review and any attached files.

Do you want your identity to be public for this peer review? For information about this choice, including consent withdrawal, please see our Privacy Policy.

Reviewer #1: No

---

## [Decision Letter · Decision Letter 1]

26 Apr 2023

PONE-D-23-05315R1The effects of GLP-1 receptor agonists on visceral fat and liver ectopic fat in an adult population with or without diabetes and nonalcoholic fatty liver disease: a systematic review and meta-analysisPLOS ONE

Dear Dr. Li,

Thank you for submitting your manuscript to PLOS ONE. After careful consideration, we feel that it has merit but does not fully meet PLOS ONE’s publication criteria as it currently stands. Therefore, we invite you to submit a revised version of the manuscript that addresses the points raised during the review process.

We look forward to receiving your revised manuscript.

Kind regards,

Tariq Jamal Siddiqi

Academic Editor

PLOS ONE

Journal Requirements:

Reviewers' comments:

Reviewer's Responses to Questions

**Comments to the Author**

1. If the authors have adequately addressed your comments raised in a previous round of review and you feel that this manuscript is now acceptable for publication, you may indicate that here to bypass the “Comments to the Author” section, enter your conflict of interest statement in the “Confidential to Editor” section, and submit your "Accept" recommendation.

Reviewer #1: All comments have been addressed

2. Is the manuscript technically sound, and do the data support the conclusions?

Reviewer #1: Yes

3. Has the statistical analysis been performed appropriately and rigorously? 

Reviewer #1: Yes

4. Have the authors made all data underlying the findings in their manuscript fully available?

Reviewer #1: Yes

5. Is the manuscript presented in an intelligible fashion and written in standard English?

Reviewer #1: Yes

6. Review Comments to the Author

Reviewer #1: I appreciate the authors for their attention to my comments and making necessary revisions. However, the clinical implications of the findings of this meta-analysis are still not effectively described. The discussion states that “The emergence of GLP-1 RA drugs provides us with another possibility for long-term drug treatment to normalize body weight, visceral fat, and hepatic ectopic fat. In the United States, GLP-1 receptor agonistsGLP-1 RAs such as liraglutide and semaglutide have been licensed by the US Food and Drug Administration (FDA) for the treatment of obesity and excessive weight, regardless of whether the patient has diabetes.” indicating that GLP-1 RAs are already being used to treat obesity. The authors should describe how the findings of this meta-analysis can be used to further improve patient care or outcomes. Moreover, the clinical implications described in the conclusion should instead be placed in the discussion in a distinct paragraph for ‘clinical implications’.

7. PLOS authors have the option to publish the peer review history of their article (what does this mean?). If published, this will include your full peer review and any attached files.

Reviewer #1: No

---

## [Author Response · Author response to Decision Letter 1]

6 Jun 2023

PONE-D-23-05315R1

The effects of GLP-1 receptor agonists on visceral fat and liver ectopic fat in an adult population with or without diabetes and nonalcoholic fatty liver disease: a systematic review and meta-analysis

PLOS ONE

Dear Dr. Li,

Thank you for submitting your manuscript to PLOS ONE. After careful consideration, we feel that it has merit but does not fully meet PLOS ONE’s publication criteria as it currently stands. Therefore, we invite you to submit a revised version of the manuscript that addresses the points raised during the review process.

A rebuttal letter that responds to each point raised by the academic editor and reviewer(s). You should upload this letter as a separate file labeled 'Response to Reviewers'.

A marked-up copy of your manuscript that highlights changes made to the original version. You should upload this as a separate file labeled 'Revised Manuscript with Track Changes'.

An unmarked version of your revised paper without tracked changes. You should upload this as a separate file labeled 'Manuscript'.

We look forward to receiving your revised manuscript.

Kind regards,

Tariq Jamal Siddiqi

Academic Editor

PLOS ONE

Journal Requirements:

Response: Thank you for your critical comments. We checked the references and there were no papers that have been retracted.

Reviewers' comments:

Reviewer's Responses to Questions

Comments to the Author

1. If the authors have adequately addressed your comments raised in a previous round of review and you feel that this manuscript is now acceptable for publication, you may indicate that here to bypass the “Comments to the Author” section, enter your conflict of interest statement in the “Confidential to Editor” section, and submit your "Accept" recommendation.

Reviewer #1: All comments have been addressed

2. Is the manuscript technically sound, and do the data support the conclusions?

Reviewer #1: Yes

3. Has the statistical analysis been performed appropriately and rigorously?

Reviewer #1: Yes

4. Have the authors made all data underlying the findings in their manuscript fully available?

Reviewer #1: Yes

5. Is the manuscript presented in an intelligible fashion and written in standard English?

Reviewer #1: Yes

6. Review Comments to the Author

Reviewer #1: I appreciate the authors for their attention to my comments and making necessary revisions. However, the clinical implications of the findings of this meta-analysis are still not effectively described. The discussion states that “The emergence of GLP-1 RA drugs provides us with another possibility for long-term drug treatment to normalize body weight, visceral fat, and hepatic ectopic fat. In the United States, GLP-1 receptor agonists GLP-1 RAs such as liraglutide and semaglutide have been licensed by the US Food and Drug Administration (FDA) for the treatment of obesity and excessive weight, regardless of whether the patient has diabetes.” indicating that GLP-1 RAs are already being used to treat obesity. The authors should describe how the findings of this meta-analysis can be used to further improve patient care or outcomes. Moreover, the clinical implications described in the conclusion should instead be placed in the discussion in a distinct paragraph for ‘clinical implications’.

Response: Thank you for your helpful comments. We added the clinical application in the discussion. 

7. PLOS authors have the option to publish the peer review history of their article (what does this mean?). If published, this will include your full peer review and any attached files.

Do you want your identity to be public for this peer review? For information about this choice, including consent withdrawal, please see our Privacy Policy.

Reviewer #1: No

---

## [Decision Letter · Decision Letter 2]

24 Jul 2023

The effects of GLP-1 receptor agonists on visceral fat and liver ectopic fat in an adult population with or without diabetes and nonalcoholic fatty liver disease: a systematic review and meta-analysis

PONE-D-23-05315R2

Dear Dr. Li,

We’re pleased to inform you that your manuscript has been judged scientifically suitable for publication and will be formally accepted for publication once it meets all outstanding technical requirements.

Kind regards,

Tariq Jamal Siddiqi

Academic Editor

PLOS ONE

Additional Editor Comments (optional):

Reviewers' comments:

Reviewer's Responses to Questions

**Comments to the Author**

1. If the authors have adequately addressed your comments raised in a previous round of review and you feel that this manuscript is now acceptable for publication, you may indicate that here to bypass the “Comments to the Author” section, enter your conflict of interest statement in the “Confidential to Editor” section, and submit your "Accept" recommendation.

Reviewer #1: All comments have been addressed

2. Is the manuscript technically sound, and do the data support the conclusions?

Reviewer #1: Yes

3. Has the statistical analysis been performed appropriately and rigorously? 

Reviewer #1: Yes

4. Have the authors made all data underlying the findings in their manuscript fully available?

Reviewer #1: Yes

5. Is the manuscript presented in an intelligible fashion and written in standard English?

Reviewer #1: Yes

6. Review Comments to the Author

Reviewer #1: I commend the authors for addressing all the comments. The revisions made by the authors have substantially improved the quality and scientific merit of the meta-analysis and I recommend acceptance of this manuscript for publication.

7. PLOS authors have the option to publish the peer review history of their article (what does this mean?). If published, this will include your full peer review and any attached files.

Reviewer #1: No

---

## [Editor Report · Acceptance letter]

16 Aug 2023

PONE-D-23-05315R2 

The effects of GLP-1 receptor agonists on visceral fat and liver ectopic fat in an adult population with or without diabetes and nonalcoholic fatty liver disease: a systematic review and meta-analysis 

Dear Dr. Li:

I'm pleased to inform you that your manuscript has been deemed suitable for publication in PLOS ONE. Congratulations! Your manuscript is now with our production department. 

Kind regards, 

on behalf of

Dr. Tariq Jamal Siddiqi 

Academic Editor

PLOS ONE